# Exploiting Semantic Coherence To Improve Prediction In Satellite Scene Image Analysis: Application to Disease Density Estimation

## Abstract

High intra-class diversity and inter-class similarity is a characteristic of remote sensing scene image data sets currently posing significant difficulty for deep learning algorithms on classification tasks. To improve accuracy, post-classification methods have been proposed for smoothing results of model predictions. However, those approaches require an additional neural network to perform the smoothing operation, which adds overhead to the task. We propose an approach that involves learning deep features directly over neighboring scene images without requiring use of a cleanup model. Our approach utilizes a siamese network to improve the discriminative power of convolutional neural networks on a pair of neighboring scene images. It then exploits semantic coherence between this pair to enrich the feature vector of the image for which we want to predict a label. Empirical results show that this approach provides a viable alternative to existing methods. For example, our model improved prediction accuracy by 1 percentage point and dropped the mean squared error value by 0.02 over the baseline, on a disease density estimation task. These performance gains are comparable with results from existing post-classification methods, moreover without implementation overheads.

## 1 Introduction

Remote sensing scene image analysis is emerging as an important area of research for application of deep learning algorithms. Application areas include land-use land-cover analysis, urban planning, and natural disaster detection. A deep learning task for labeling a scene image is typically formulated as conditional probability of the form in Eq. 1 Liu et al. (2019),Albert et al. (2017), Nogueira et al. (2016), Castelluccio et al. (2015), Mnih (2013), Mnih & Hinton (2010),

$$P(l_i|s_i) \tag{1}$$

where $l_i$ is label for image patch $s_i$. This formulation is sufficient for problems where spatial situatedness of a scene, which embodies knowledge of semantic likeness between neighborhoods in the geophysical world, is not important. However, for problems which require knowledge of neighborhood the formulation in Eq. 1 becomes inadequate. An example of such a problem would be estimating disease density for a small geographical region of interest, in which case the probability of label $l$ is likely to depend on the labels for neighboring regions due to semantic coherence among them.

The problem of how to improve model prediction by leveraging semantic coherence among neighboring scene images has previously been considered in the literature. Previous studies consider the problem as a post-classification task. For example, Bischof et al. (1992) used a second classifier to do pixel smoothing to refine predictions made by another classifier. Based on a 5x5 window, a filter assigns pixels to the majority class if it had been assigned a different class. In Mnih (2013), a post-processing architecture is suggested for incorporating structure into image patch prediction. It involves stacking neural networks (NN) such that the output from a previous one becomes input for the next. Idea is for each network to clean up predictions of previous one in order to progressively

improve overall accuracy. While improved model performance was achieved by these methods, they have overhead of performing same classification task in at least two stages. In other words, you need a minimum of two NN to perform the same classification task.

Unlike post-classification methods, this work considers the problem of improving model accuracy on scene images by exploiting knowledge of neighboring scenes as part of the model training process. We make the assumption that $l$ is conditionally co-dependent on information embedded in scene image $i$ and in other similar, neighboring image $j$ such that the problem is formulated as probability of the form in Eq. 2,

$$P(\boldsymbol{l}_i|\boldsymbol{S}_i, \boldsymbol{S}_j) = \Pi_{i=1}^{m} P(l_i|s_i, s_j) \qquad (2)$$

where $s_j$ is image for a neighboring tile that is most similar to index tile $i$ and $P(\boldsymbol{l}_i|\boldsymbol{S}_i, \boldsymbol{S}_j)$ is observed probability distribution.

We used Convolutional Neural Networks (CNN) for modeling the observed probability distribution in Eq. 2. A network architecture is proposed for training our model consisting of four components: a siamese sub-network, a similarity metric learning component, a convolutional network, and a decision layer. The siamese sub-network takes two neighboring scene images as input and extracts features from each. The similarity learning component evaluates how similar the input images are, using the extracted features. If the two input images are found to be similar the convolutional network learns additional features based on the merged feature vector, otherwise those from the index tile are used alone. We implemented the decision layer to perform classification or regression. A baseline model was implemented that takes a single image, the index tile $i$, as input.

Empirical results show the proposed model consistently outperforms the baseline. In addition to improving predictive performance with a relatively small training set, our model is fast to train since it uses a pre-trained model for the siamese sub-network. Furthermore, it does not require another NN to smooth out its predictions as is the case with post-classification approaches, while achieving comparable performance gain. In summary, our contributions include the following.

1. We propose an approach for training a probabilistic deep learning model to improve prediction accuracy by exploiting semantic coherence between neighboring tiles in aerial scene images. A CNN architecture is suggested for this purpose.

2. We provide empirical evidence that demonstrates the viability of this approach on a disease density estimation task.

3. Lastly, we discovered an important limitation of the synthetic minority over-sampling technique (SMOTE). This method fails when used for oversampling an under-represented class whereby knowledge of spatial proximity between scene image data points must be preserved, an important requirement under the framework of learning deep features over neighboring scene images introduced in this work.

## 2 RELATED WORK

Remote sensing scene images represent a difficult problem for deep learning algorithms due to high intra-class diversity and inter-class similarity (Cheng et al., 2017). Previous efforts to address this problem is presented first, followed by recent use of siamese networks to improve discriminative ability of CNN.

### 2.1 IMPROVING MODEL ACCURACY BY EXPLOITING COHERENCE IN SCENE IMAGES

One of the earliest works aimed at improving model accuracy by exploiting semantic coherence between neighboring images is by Bischof et al. (1992). The problem addressed there is 'salt and pepper' noise in classified satellite images. The authors employ a multi-layer perceptron (MLP) consisting of five hidden units to classify low resolution (30m) satellite scene images. Their proposed network has four output units corresponding to four land-use classes. The sigmoid activation function was used with both hidden and output units. The network had seven input units, each corresponding to one of seven Landsat TM spectral bands. It generates two separate channels when

classifying an input image: one for class prediction result and the other representing how confident the classification is. A post-classification smoothing operation is then performed using a two-layer NN whose input is the classified image within a 5 x 5 window. For each pixel, the predicted class and confidence information are used as input to decide its new class belonging by applying a majority filter. Gain in classification accuracy was 2.9 percent for the network whose input data was enhanced with texture information (and 4.9 percent for the one without enhancement).

Mnih (2013) introduce indirect dependencies between outputs of a model by using knowledge of structure (e.g. shape) to resolve the issue of disconnected blobs and holes or gaps in predicted building and road network maps. The architecture used consists of stacking NN, each using as input the outputs of a previous network. Precisely, let $\hat{M}_0$ be the map predicted by a model. The $i$th level cleanup NN ($f_i$) takes a $w_s$ x $w_s$ patch of $\hat{M}_{i-1}$ and outputs a $w_m$ x $w_m$ patch of $\hat{M}_i$. The model $f_i$ is trained by minimizing the negative log likelihood on patches of the observed map $\hat{M}$, just like other NN in the architecture. Thus, the $i$th cleanup NN improves predictions of the previous NN $f_{i-1}$. The authors extended this idea to Conditional Random Fields (CRF) i.e. pairwise lattice CRF to introduce explicit dependencies between pairs of neighboring pixels. For both NN and CRF, an average gain of 0.0195 was achieved in precision and recall break-even points.

The above works demonstrate that post-classification smoothing operations can improve performance of deep learning models. However, a draw back of those approaches is the necessary requirement to implement additional models to perform smoothing operations. Therefore, approaches that sidestep this requirement would be a welcome alternative.

## 2.2 SIAMESE NEURAL NETWORKS

Another innovation that inspired the current work is siamese neural networks (SNN) introduced by Bromley et al. (1993) and Baldi & Chauvin (1993). A SNN architecture consists of two identical (shared weight) sub-networks merged by the same function at their outputs. During training each sub-network extracts features from one of two concurrent inputs, after which the joining function evaluates whether the input pair is similar based on a distance metric, such as cosine of angle between the pair of feature vectors. Output from the SNN is a decision score e.g., 1 for similar image pair, 0 otherwise.

SNN have recently been used for remote sensing scene image understanding. Liu et al. (2019) used a siamese architecture to improve discriminative ability of CNN for satellite scene image classification. The architecture consists of two identical CNN models, 3 additional convolutional layers, and one square non-parametric layer. Each branch of the siamese network is responsible for extracting features from each dual input image, giving feature vectors $f_1$ and $f_2$. Two of the convolutional layers take these feature vectors as input and perform additional learning before a softmax function predicts the label for each input image. On the other hand, the square layer takes $f_1$ and $f_2$ as input and outputs the tensor $f_s = (f_1 - f_2)^2$ which is used to measure similarity between the input pair. The model thus produces three outputs: predicted label for each input image and a score for their similarity.

More precisely, feature discrimination enhancement is achieved in the square layer by imposing a metric learning regularization term that minimizes Euclidean distance between a similar image pair and maximizes that for a dissimilar pair (Eq. 3),

$$D(x_i, x_j) = ||f_1 - f_2||_2^2 \tag{3}$$

where $D(x_i, x_j)$ is distance between a pair of input images. A margin $\tau$ is set to separate as far apart as possible, the similar image pairs from dissimilar ones in feature space. If $(x_i, x_j)$ is from the same scene class the distance between them is less than $\tau$, otherwise it is greater than $\tau$. Let label for training pair $(x_i, x_j)$ be $(y_i, y_j)$, respectively. The formulation for Euclidean distance between them is given by (Eq. 4),

$$\begin{cases} D(x_i, x_j) < \tau, y_i = y_j \\ D(x_i, x_j) > \tau, y_i \neq y_j \end{cases} \tag{4}$$

This model only minimizes feature distance between similar input pairs through the margin $\tau$ to achieve feature discrimination.

Therefore, the model in Liu et al. (2019) optimizes for two objective functions: a distance loss function and a regularization term as shown in Eq. 5,

$$Dist(f_1, f_2) = \sum_{i,j} D(x_i, x_j), \ D(x_i, x_j) < \tau \qquad (5)$$

Of the three pre-trained models experimented with (AlexNet, VGG-16, and ResNet-50) highest feature discrimination accuracy gain (1.14 percentage point) was achieved with a Siamese AlexNet model on the challenging NWPU-RESIC45 (Cheng et al., 2017) benchmark data set. Cheng et al. (2018) reported a 1.53 percentage point gain in accuracy on the same data set.

## 3 LEARNING DEEP FEATURES OVER NEIGHBORING SCENE IMAGES

### 3.1 OVERVIEW

An end-to-end pipeline is proposed consisting of four components: a siamese sub-network, similarity metric learning function, convolutional network, and decision layer. The siamese sub-network takes a pair of scene images $s_i$ (tile we want to predict a label for) and $s_j$ (other tile neighboring $s_i$) as input and extracts features from each, outputting feature vectors $f_i$ and $f_j$, respectively. The similarity function takes as input the two feature vectors and learns a similarity metric between them. If the feature vector pair $(f_i, f_j)$ is found to be similar they are merged and the resulting feature vector used to train the convolutional network, otherwise training is done using $(f_i)$ alone. Output from the convolutional network is passed as input to the decision layer to predict a label (or to regress a numerical value) for image patch $s_i$. The proposed and baseline architectures are shown in Figures 1a and 1b, respectively.

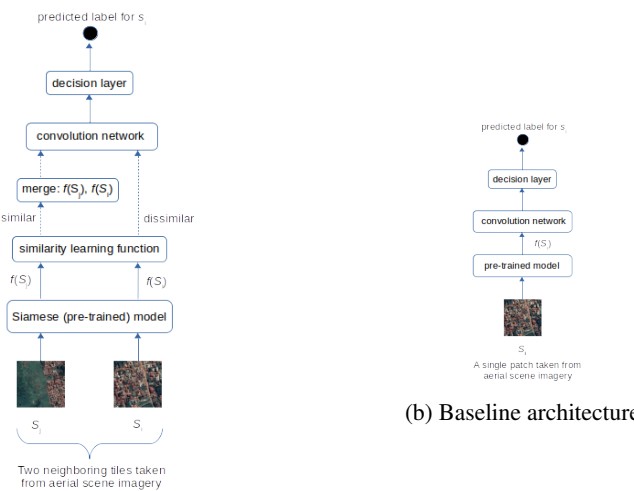

(a) Proposed network architecture.

(b) Baseline architecture.

Figure 1: Network architectures used in this work.

### 3.2 ARCHITECTURE DESCRIPTION

#### 3.2.1 SIAMESE SUB-NETWORK

To implement the siamese sub-network we explored two state-of-the-art pre-trained CNN models i.e. Xception net Chollet (2016) and ResNet-50 He et al. (2016) in a transfer learning (fine-tuning) strategy since we had a small training set. The architecture of Xception model is based on separating learning of space-wise and channel-wise feature representations. The result is better performance

over architectures such as Inception V3, ResNet-152, and VGG-16 in terms of computational speed and classification accuracy. On the other hand, ResNet architectures address the problem of vanishing/exploding gradients in deep neural network architectures by inserting shortcut connections between convolutional blocks. The ResNet model has benefit of enabling deeper architectures to be utilized in CNN models since these provide higher accuracy rates than shallow architectures.

### 3.2.2 SIMILARITY METRIC LEARNING

The similarity function takes as input feature vector $(f_i, f_j)$ to evaluate how similar the input image pair is. This unit outputs one of a binary decision score i.e neighboring input pair $(s_i, s_j)$ is similar (1) or dissimilar (0). The optimization function used here works by minimizing feature distance between similar images. Following work in Liu et al. (2019), given a pair of input image $s_i$ and $s_j$ the distance between them is calculated using Eq. 3. A margin $\tau$ separates similar image features from dissimilar ones such that $D(f_i, f_j) < \tau$ for a similar pair, $D(f_i, f_j) > \tau$ otherwise. We also optimized for two objective functions as defined in Eq. 5.

An important question to consider is how to identify a neighboring image that is semantically closest in likeness to index tile $i$, over which to co-learn features. For any image patch in the middle region of a $w$ x $h$ scene image $S$ where w$\geq$3 and h$\geq$3, there will be eight such tiles to choose from, considering four adjacent and diagonal neighbors. For edge tiles there will be five while for corner tiles there will be three neighbors, Figure 2 in Appendix.

We can determine neighbors of the tile at location $i$ by specifying the maximum Harversine distance within which the centroid coordinates of all neighbors will lie. Take as an example tiles of size $250m^2$ in a 3 x 3 scene image. The centroid of all eight neighbors of the center tile would fall within a maximum distance of 350m from its own centroid. We can evaluate how similar tile at $i$ is to its neighbor at location $j$ using Euclidean distance. If at least one is similar, we select the neighbor with least distance to be most similar. Given that $D(s_i, s_j) < \tau$ for images from same scene class, a neighboring image is most similar to $s_i$ if it has the smallest distance Eq. 6,

$$\hat{s_i} = \arg \min_{s_i}(s_i, s_j) \qquad (6)$$

where $\hat{s_i}$ is neighboring image most similar to index tile at location $i$.

In the current analysis however, we consider only the preceding neighbor to tile at location $i$ (image $s_{i-1}$) for similarity analysis. The loss function used is binary cross-entropy, defined in Eq. 7.

$$L(W, b) = -\sum_{i=1}^{N} y_i \times \log \hat{y_i} + \left(1 - y_i\right) \times \log \left(1 - \hat{y_i}\right) \qquad (7)$$

### 3.2.3 CONVOLUTIONAL NETWORK

This network is trained using features resulting from merging feature vector pair $(f_i, f_j)$ if respective input image pair $(s_i, s_j)$ is found to be similar and it outputs feature vector $f_m(f_i, f_j)$. If the input pair is dissimilar, only the feature vector $f_i$ is fed to the convolutional network, which outputs $f_i(f_i)$. Our implementation used three dense layers in the convolutional network. Merge operation used is feature averaging.

### 3.2.4 DECISION LAYER

For a classification task we learn the parameters (weights) of the CNN by minimizing the negative log likelihood of the training data under our model. For the multi-class problem considered in this work the negative log likelihood under the model in Eq. 2 assumes the form of a cross-entropy between the probability distribution for observed labels $l$ and the predicted label probabilities $\hat{y}$ (Eq. 8),

$$L(W, b) = -\sum_{i=1}^{N} \sum_{j=1}^{M} y_i \times \log \hat{y_i} \qquad (8)$$

where M, the number of classes, is 3.

The outer sum of objective function $L$ is over all training samples. Stochastic gradient descent with mini-batches is used for optimizing $L$. Our implementation for the current problem uses a fully connected network (FCN) with a 3-way softmax classifier.

Taken as a regression task, we optimized for the mean absolute error (MAE) loss and mean squared error (MSE). The form of MAE we used is given by Eq. 9,

$$L(W, b) = \frac{1}{2N} \sum_{i=1}^{N} \sum_{j=1}^{M} \left| \hat{y}_i - y_i \right| \tag{9}$$

where M, the number of features to be predicted, is 2. For MSE the loss function is given by Eq. 10,

$$L(W, b) = \frac{1}{N} \sum_{i=1}^{N} \frac{1}{M} \sum_{i=1}^{M} \left( \hat{y}_i - y_i \right)^2 \tag{10}$$

## 4 EXPERIMENTS

We applied our method to the task of estimating disease density from satellite scene imagery. Below we describe data sets and methods used.

### 4.1 METHODS AND DATASET

**Dataset**

A number of data sets were used in our experiments. The epidemic data consists of monthly disease case counts aggregated by sub-county for year 2015. We dis-aggregated the data to $250m^2$ grids, our geospatial unit of analysis, using population data from Facebook & CIESIN (2016). The latter data gives estimate of people living in a $30m^2$ grid based on recent census data and high resolution ($0.5m^2$) satellite imagery. The data was up-sampled to $250m^2$ grid to be consistent with our spatial unit of analysis. The population data was used to create a weighting scheme for dis-aggregating epidemic data. The building (housing) concentration data consists of satellite imagery extracted from Google Static Maps API using method in Sanya & Mwebaze (2018). It consists of land-use type buildings i.e. objects of interest in a satellite scene image is buildings or houses. The housing data was used as input to train the model, as proxy for indoor overcrowding - a known risk factor for our case study disease. All input data was pre-processed into feature vectors and normalized to (0,1) value range by min-max scaling or (-1,1) by standardizing (zero mean, unit variance).

**Methods**

We created disease density classes out of the epidemic data to make it a classification task. We did this by binning the normalized data such that each $250m^2$ grid assumed a class value depending on which bin its disease density lies in. Following the procedure in Robinson et al. (2017) for binning population data, we created a matrix **C** where an entry $C_i = 0$ if $5.74e - 05 \leq d_i \leq 1.52e - 4$, 1 if $1.52e - 4 < d_i \leq 2.46e - 4$, and 2 if $2.46e - 4 < d_i \leq 3.4e - 4$, where $d_i$ is normalized disease density. The classes correspond to the semantic labels 0, 1, 2 for high, low, and moderate disease density with 3,628, 6,472, and 1,970 membership count, respectively.

Let $D$ be a grid of disease case counts for a region of interest at time $t$, $C$ grid of disease density class label values, and $S$ grid of housing concentration data. For every disease case count value $d_i$ and density class label value $c_i$ there is an associated housing concentration image $s_i$.

We formulated the learning task as estimating unknown function $f$ based on the idea of learning deep features over neighboring scene images (one neighbor in this case) using Eq. 11 (Figure 1a),

$$c_i = \begin{cases} f(s_i, s_j) & \text{if} \quad D(s_i, s_j) < \tau \quad \text{and} \quad \arg\min_{s_i}(D(s_i, s_j)), \\ f(s_i) & \text{if} \quad D(s_i, s_j) > \tau. \end{cases} \tag{11}$$

Table 1: Overall accuracy, precision, recall, and F1-score for similarity metric learning

|  | Accuracy | Precision | Recall |
|---|---|---|---|
| ResNet-50 | 57 | 0.39 | 0.10 |
| Xception | 83 | 0.70 | 0.60 |

Table 2: Overall accuracy, precision, recall, F1-score for baseline and proposed model.

| | Accuracy | | Precision | | Recall | | F1-score | |
|---|---|---|---|---|---|---|---|---|
| | Bline | Model | Bline | Model | Bline | Model | Bline | Model |
| ResNet-50 | **44** | 42 | 0.0 | **0.11** | 0.0 | **0.3** | 0.0 | **0.5** |
| Xception | 33 | **34** | 0.13 | **0.15** | 0.26 | **0.39** | 0.18 | **0.22** |

where $c_i$ is estimated disease density for a $250m^2$ patch of land on earth's surface (approx. $224m^2$ pixels for an image), represented by image $s_i$. $s_j$ is image of a neighboring scene that is most similar to image of tile at location $i$, identified using method in Eq. 6.

We used CNN to estimate function $f$ because the mapping from input data to disease incidence estimate is non-linear, noisy, and dependent on semantic content of input data. To train our model we used transfer learning by fine-tuning two pre-trained models on our data set i.e., Xception net and ResNet-50. Transfer learning eliminates network architecture design time while often achieving higher accuracy rates than training a model from scratch.

We applied the same regularization methods and other hyperparameter values (Table 4 in Appendix) to ensure uniformity across the proposed and baseline network architectures. Once a satisfactory model was achieved, it was trained on a combined data set (12,070 images) of training (9,656) and validation (2,414) before evaluating on the test set (1,189).

## 4.2 RESULTS

### 4.2.1 SIMILARITY METRIC LEARNING

Evaluation results for similarity metric learning for our proposed model built on top of ResNet-50 and Xception net are shown in Table 1. ResNet-50 performed better than Xception net on precision (0.39 vs. 0.70) and recall (0.10 vs. 0.60) on the task of detecting whether or not a pair of satellite scene images is similar, even though the overall performance is low for both pre-trained models (maximum OA score of 83 percent got with Xception net).

### 4.2.2 CLASSIFICATION PERFORMANCE

Performance results for the baseline and proposed model are shown in Table 2. Overall, the Xception based model performed better than the ResNet-50 based one on the task of estimating disease density from housing satellite scene data when performance is measured using precision, recall, and F1-Score metrics. However, both models generally performed poorly, achieving a maximum overall accuracy score of 44 percent for ResNet-50. Our proposed model however, consistently out performed the baseline across both pre-trained models (scores marked with bold). For example, using Xception net as the Siamese network our model achieved precision score of 0.15 vs. 0.13, recall of 0.39 vs. 0.26, and f1-score of 0.22 vs. 0.18 for the baseline model, respectively. A similar performance trend is observed for the model based on ResNet-50.

Confusion matrix plots for baseline and our model built using Xception net are shown in Figure 3a and 3b, respectively (Appendix). On the other hand Figure 4 shows AUROC plots. Both show that our model performed better than the baseline on medium and low disease density classes with 58 vs. 48 percent and 47 vs. 37 percent, respectively. However, the model performed worse than the

baseline on high disease density class (0 vs. 13 percent, respectively). Generally, the results from both models are as good as chance.

### 4.2.3 REGRESSION PERFORMANCE

Table 3 gives MAE and MSE scores for baseline and proposed model. Overall, the model built using ResNet-50 performed better than the one built with Xception net when used to estimate disease density from housing concentration satellite scene image data. Again, our proposed model performed consistently (even though marginally) better than the baseline (scores marked with bold). It achieved MAE score of 0.35 vs. 0.38 and MSE score of 0.18 vs. 0.20 for the baseline, respectively. A similar trend is observed for the model built with Xception net as base.

Table 3: MAE and MSE scores for baseline and proposed model.

| | MAE | | MSE | |
| | Baseline | Model | Baseline | Model |
|---|---|---|---|---|
| ResNet-50 | 0.38 | **0.35** | 0.20 | **0.18** |
| Xception | 0.39 | **0.37** | 0.20 | **0.19** |

## 5 DISCUSSION

Our model performed better than the baseline in both the classification and regression tasks for disease density estimation. For example, our model achieved 1 percentage point gain in accuracy over the baseline model. While a gain as result of deploying a siamese network to boost discriminative power of CNN for aerial scene image classification is consistent with findings in previous studies overall results from our model are poor. For instance, our model was only able to attain a maximum overall accuracy of 34 percent on the classification task. We would like to pin these poor results to a combination of three factors. First, the small data set used to train our model (12,070 images) could have impacted accuracy negatively, despite the use of regularization methods.It is therefore, possible that our model suffered the problem of overfitting. Secondly, our data set was unbalanced. It is likely that the extra parameter we introduced in the loss function to weight classes by giving higher importance to under-represented classes did not work as expected. The result could have been that our model did not learn all the necessary features required to make a prediction but rather could have resorted to guessing the output and hence, failing to generalize well over the test set. Class imbalance in our data set could also have negatively affected feature correlation which in turn could have reduced model performance. Besides, well-known methods for mitigating sample size bias in imbalanced data sets, for example by over-sampling under-represented classes Chawla et al. (2002) could not be applied directly to our data set without modifying the algorithm. That is because it was not immediately clear how to preserve spatial proximity between neighboring tiles, an idea that is central to learning deep features over neighboring scene images.However, despite the low overall performance by our model we have been able to demonstrate that it is possible to improve model accuracy by learning deep features over neighboring scene images in a disease density estimation task.

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

## 6 APPENDIX

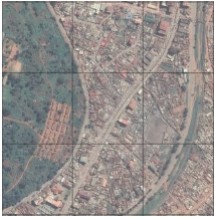

Figure 2: Finding a neighboring image $j$ that is semantically most similar to image $i$.

Table 4: Hyperparameter values used in our experiments

| Hyperparameter | Method or Value |
|---|---|
| Activation function (convolutional layer) | ReLU |
| Activation function (classification layer) | Softmax |
| Optimization method | SGD (Adadelta) |
| Loss function | Multi-class cross-entropy |
| Learning rate | 0.1 (halved every 10 epochs) |
| Batch normalization | (momentum=0.9) |
| Weight initialization | Mean=0, std=0.01 |
| Bias initialization | Mean=0.5, std=0.01 |
| Weight decay type (kernel regularizer) | L2 (0.01) |
| Amount of weight decay | 0.0002 |
| Drop out | Not used |
| Mini-batch size | 16 |
| Number of epochs | 50 |

(a) Confusion matrix plot for baseline.     (b) Confusion matrix plot for our model.

Figure 3: Confusion matrix plots for models used in this study.

(a) AUROC plot for baseline.     (b) AUROC plot for our model.

Figure 4: AUROC plots for models used in this study.

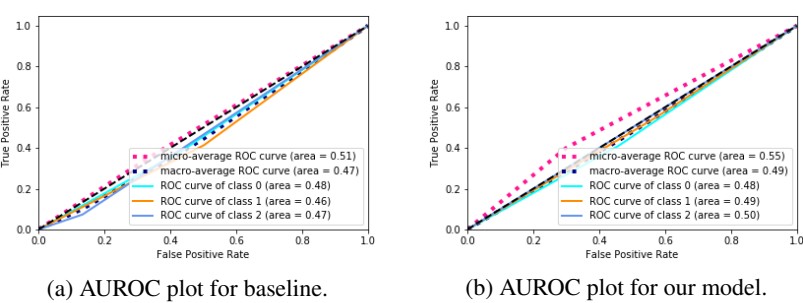

