# OpenReview forum: "EXPLOITING SEMANTIC COHERENCE TO IMPROVE PREDICTION IN SATELLITE SCENE IMAGE ANALYSIS: APPLICATION TO DISEASE DENSITY ESTIMATION"
_ICLR.cc/2020/Conference — Reject_

### Official Review · AnonReviewer2 · 2019-10-23
**Official Blind Review #2**

**Rating:** 3

**Review:**

The authors propose a method to extract features utilizing the adjacency between patches, for better classification/regression of satellite image patches. The proposed method achieves better results compared to a straightforward baseline method.


I have several significant concerns:

- In the abstract, the authors claim that existing approaches such as post-classification add computational overhead to the task, whereas the proposed method does not add significant overhead. However, to me, post-classification can be very simple and straightforward, whereas the proposed method adds a series of computations: the proposed method not only extracts features from the input image, but also for another neighboring image; then features are combined (if two images are similar), before feeding into the network. The authors need to validate the claim that their method is more efficient.

- The baseline the authors compare to is weak. There are existing works on satellite image classification/regression. Many of them also use semantic/contextual information, or aim to improve the robustness of features. For example:

[1] Derksen et al. Spatially Precise Contextual Features Based on Superpixel Neighborhoods for Land Cover Mapping with High Resolution Satellite Image Time Series. IGARSS 2018.

[2] Ghassemi et al. Learning and Adapting Robust Features for Satellite Image Segmentation on Heterogeneous Data Sets. Geoscience and Remote Sensing 2019.

I understand that the authors cannot compare to everything. But the authors should compare to representative baseline methods. Methods mentioned in the related work section (Section 2.1) can also be compared to.

- The proposed method is very application specific. The author only discussed the remote sensing application. Given the ICLR community's interest in general methods that can be applied to (or already been tested on) multiple applications, the paper would have been stronger if the methods applicabilityto other domains was discussed (and even better demonstrated).

**Experience Assessment:**

I have published one or two papers in this area.

**Review Assessment: Checking Correctness Of Derivations And Theory:**

I assessed the sensibility of the derivations and theory.

**Review Assessment: Checking Correctness Of Experiments:**

I assessed the sensibility of the experiments.

**Review Assessment: Thoroughness In Paper Reading:**

I read the paper at least twice and used my best judgement in assessing the paper.

---

> ### Author Response · Authors · 2019-11-14
> **We agree with many of the reviewer concerns**
>
> Summary of reviewer concerns:
>
> 1. Our claim of having developed an approach for improving prediction accuracy for satellite scene image analysis that has greater efficiency than post-classification approaches is not validated with experiments (reviewer #2, #5).
> 2. We should compare performance of our model against a fair baseline, like the post-classification methods cited in the paper (reviewer #2, #4).
>
> Response to concerns 1, 2 – we acknowledge this weakness. We will conduct validation experiments.
>
> 3. Our method has not been generalized to other application domains, thus limited in scope (reviewer #2, #5).
>
> Response – we will consider the possibility of generalizing to other application domains
>
> 4. Use of a hard threshold for similarity metric seems arbitrary. Suggest to take “all geographical neighborhood of a patch into account when making a prediction e.g., with a coarse-to-fine prediction approach.” That the aggregation of features can be learned and more sophisticated than average pooling (reviewer #4).
>
> Response – aggregating features through learning will be considered in the next phase of our work. Taking all neighbors of a patch is the main idea we have, though not stated in our paper. However, the analysis in our current paper is scoped for only one neighbor. We are currently designing further experiments that consider all neighbors.
>
> 5. Discussion of the paper is said to be weak (reviewer #4).
>
> Response – We will improve the discussion upon addressing concerns 1, 2, 3, and 4.
>
> We thank all reviewers of our paper for their generous feedback.

---

### Official Review · AnonReviewer4 · 2019-11-04
**Official Blind Review #4**

**Rating:** 1

**Review:**

The paper proposes to do a coupled inference over pairs of geographically close images instead of a single image for satellite imagery. The coupling is done with an average pooling of the feature vectors when the neighbouring patches are detected to be similar enough based on a threshold on the L2 distance of these features. The method is applied to tasks of estimating crowding population, and diseases density, from satellite images.

The paper have little novelty. The approach reduces to a smoothing method over pairs of neighbouring patches, that is only activated sometimes based on a hard threshold. This seems arbitrary and there are many competing approaches that could be applied.
One could think about taking all the geograpical neighbourhood of a patch into account when making a prediction, e.g. with a coarse-to-fine prediction approach; the aggregation of features can be learned and more sophisticated than average pooling. Using a single-image baseline is not fair. The discussion is not up to the level of ICLR and offers mostly guesswork.

**Experience Assessment:**

I have published one or two papers in this area.

**Review Assessment: Checking Correctness Of Derivations And Theory:**

I carefully checked the derivations and theory.

**Review Assessment: Checking Correctness Of Experiments:**

I carefully checked the experiments.

**Review Assessment: Thoroughness In Paper Reading:**

I read the paper thoroughly.

---

### Official Review · AnonReviewer5 · 2019-11-07
**Official Blind Review #5**

**Rating:** 1

**Review:**

This papers proposed a solution to the problem of disease density estimation using satellite scene images. One common challenge in this type of applications is having a high intra-class diversity and a high inter-class similarity. The solution proposed by the authors is based on the use of siamese networks to extract features from pairs of neighbouring images, and merge the features only if they are similar. The authors claim that this approach alleviate the need of a post-classification smoothing.

Advantages:
The idea of merging siamese features for similar tiles only is sound. The paper is clearly written and structured. The shown results seem to outperform the baseline.

Drawbacks:
The paper seems of fairly limited novelty. Moreover, it is centered around  one particular application. Although the task is approached with both a classification and a regression model, the classification dataset is obtained by a simple binning which makes the two tasks highly related. It would be interesting to have different settings to test the consistency of improvement with the proposed method. Finally, the authors claim that the method alleviates the need to post-classification smoothing, but this cannot be straightforwardly concluded from the conducted experiments. It would be interesting to have a more thorough comparison to other methods that use post-classification processing.

**Experience Assessment:**

I have read many papers in this area.

**Review Assessment: Checking Correctness Of Derivations And Theory:**

N/A

**Review Assessment: Checking Correctness Of Experiments:**

I carefully checked the experiments.

**Review Assessment: Thoroughness In Paper Reading:**

I read the paper at least twice and used my best judgement in assessing the paper.

---

### Decision · Program_Chairs · 2019-12-19

**Decision:**

Reject

**Comment:**

This papers proposed a solution to the problem of disease density estimation using satellite scene images.  The method combines a classification and regression task.  The reviewers were unanimous in their recommendation that the submission not be accepted to ICLR.  The main concern was a lack of methodological novelty.  The authors responded to reviewer comments, and indicated a list of improvements that still remain to be done indicating that the paper should at least go through another review cycle.